# Macrophages-induced IL-18–mediated eosinophilia promotes characteristics of pancreatic malignancy

Hemanth Kumar Kandikattu ⓘ, Murli Manohar, Alok Kumar Verma, Sandeep Kumar, Chandra Sekhar Yadavalli ⓘ, Sathisha Upparahalli Venkateshaiah ⓘ, Anil Mishra ⓘ

Reports indicate that accumulated macrophages in the pancreas are responsible for promoting the pathogenesis of chronic pancreatitis (CP). Recently, macrophage-secreted cytokines have been implicated in promoting pancreatic acinar-to-ductal metaplasia (ADM). This study aims to establish the role of accumulated macrophage-activated NLRP3-IL-18-eosinophil mechanistic pathway in promoting several characteristics of pancreatic malignancy in CP. We report that in a murine model of pancreatic cancer (PC), accumulated macrophages are the source of NLRP3-regulated IL-18, which promotes eosinophilic inflammation-mediated accumulation to peri-ductal mucin and collagen, including the formation of ADM, pancreatic intraepithelial neoplasia (PanINs), and intraductal papillary mucinous neoplasm. Most importantly, we show improved malignant characteristics with reduced levels of oncogenes in an anti–IL-18 neutralized and IL-18 gene deficient murine model of CP. Last, human biopsies validated that NLRP3-IL-18–induced eosinophils accumulate near the ducts, showing PanINs formation in PC. Taken together, we present the evidence on the role of IL-18–induced eosinophilia in the development of PC phenotype like ADM, PanINs, and ductal cell differentiation in inflammation-induced CP.

## Introduction

We have reported that inflammatory macrophages accumulate in cerulein-induced chronic pancreatitis (CP) (Manohar et al, 2018a), and a recent report suggested that macrophage-secreted cytokines drive pancreatic acinar-to-ductal metaplasia (ADM) (Liou et al, 2013). However, it is not yet clear what macrophage-derived specific mediators or mediator-induced responses are involved in promoting pancreatic disorders including ADM. Pancreatic disorders associated with tissue eosinophilia include pancreatic cancer (PC), autoimmune pancreatitis (AIP), eosinophilic pancreatitis (EP), and CP (Sah et al, 2010; Manohar et al 2017a, 2017b). CP and EP are fibro-inflammatory disorders (Manohar et al 2017a, 2018b), and symptoms include abdominal pain, vomiting, diarrhea, and other gastrointestinal symptoms with marked eosinophilic infiltration in

pathological samples (despite the lack of a standard for eosinophil counts) and no organ involvement outside the digestive system (Manohar et al 2017a, 2020). Eosinophils also play a key role in food-induced allergic responses (Suzuki et al, 2012; Tse & Christiansen 2015). Blood and tissue eosinophilia with marked degranulation have been reported in a number of allergic diseases associated with food (Shukla et al, 2015; Kandikattu et al, 2019). EP is a relatively rare disease characterized by local or diffuse infiltration of eosinophils into the pancreas (Tian et al, 2016; Manohar et al, 2021). EP is easily misdiagnosed as PC because of similarities in their clinical symptoms, and some cases of PC are associated with eosinophilia (Euscher et al, 2000). Interestingly, based on clinical and experimental evidence, IL-18 seems to play an important role in the pathogenesis of chronic and EP (Yamaguchi et al, 1995; Abraham et al, 2003; Manohar et al 2017b, 2018b). Previously, we showed that IL-18 has a critical role in the generation, maturation, and transformation of naïve eosinophils to pathogenic eosinophils (Venkateshaiah et al, 2018). Eosinophils are the source of the profibrotic cytokine TGF-$\beta$ and are involved via the signaling molecule SMAD4 in promoting pancreatic fibrosis that may lead to the development of characteristic features observed in PC (Thakur et al, 2016; Ahmed et al, 2017; Kandikattu et al, 2021a).

Patients suffering from CP carry a significantly higher risk of developing PC (Shimosegawa et al, 2009). Several experimental models have provided evidence of the presence of eosinophils in PC; however, these models failed to reveal the mechanistic pathway involved in the induction of pancreatic eosinophilia and the role of eosinophils in the initiation and progression of pancreatic carcinoma. Thus, a novel experimental model is required that provides a stepwise progression of the mechanistic events occurring in the development of eosinophilic inflammation-associated features observed in PC like acinar cell hypertrophy, ADM, and pancreatic intraepithelial neoplasia (PanIN). Herein, we present a murine model of chronic EP following the intraperitoneal administration of cerulein and azoxymethane (AOM) in mice. AOM, a gene mutation agent, has been earlier used to develop inflammation-mediated cancer characteristics in laboratory animals (Orii et al, 2003); therefore, we used AOM in addition to cerulein, which is commonly used to promote pancreatitis in mice. This murine model

Department of Medicine, Tulane Eosinophilic Disorders Centre, Section of Pulmonary Diseases, School of Medicine, Tulane University, New Orleans, LA, USA

Correspondence: amishra@tulane.edu

mechanistically showed NLRP3-regulated IL-18-induced eosinophil accumulation and degranulation, merged pancreatic ducts, pancreatic intraepithelial neoplasia 1 (PanIN1), PanIN2, PanIN3, and intraductal papillary mucinous neoplasm (IPMN). Induced IL-18 is reported in CP (Manohar et al, 2018b) and PC, including pancreatic ductal adenocarcinoma (PDAC), and is correlated with a poor survival rate (Carbone et al, 2009). Duct merging and formation of PanINs and IPMN are characteristic features of human malignant neoplasm Jaidev et al, 2021. The current study defines the significance of macrophage-induced IL-18–mediated pancreatic eosinophilia-associated inflammatory responses are critical in the development of pathological features of PC that progresses into malignancy in inflammation-mediated CP.

# Results

### Establish an eosinophilic inflammation-mediated murine model of CP that shows pathological features that mimic PC

We aimed to develop a murine model of CP that provides a mechanistic understanding of the development of several characteristic features observed in human PC. Accordingly, an experimental model was developed by delivering an intraperitoneal injection regimen of three AOM and eight cerulein treatments over a total of 19 wk following the protocol shown in Fig 1A. The gross morphology of the pancreas at the end of the treatment regimen showed decreased tissue mass with calcified tissue spots resembling moderate tumor growth in cerulein-with-AOM–treated mice, whereas atrophic pancreas was observed in mice treated with cerulein only, and saline-treated mice and mice administered AOM alone showed normal pancreas pathology (Fig 1Bi–iv). Furthermore, tissue sections of mice treated with saline alone and with AOM alone histologically showed normal acinar cell, ductal cell, and islet cell morphology, whereas cerulein-treated mice showed hypertrophic acinar cells and ductal cells with accumulated inflammatory cells (Fig 1Ci–iii). In addition, several human PDAC characteristics including a merging of the ductal cells, PanIN1, PanIN2, and PanIN3 (Fig 1Civ) with thick periductal stroma (Fig 1Di–iii), and ADM were observed (Fig 1Div) in cerulein-with-AOM–treated mice. Furthermore, the formation of IPMN (Fig 1Ei), mucinous cystic neoplasm (MCN) (Fig 1Eii), and induced mucin secretion in the tissue section and around the pancreatic ducts (Fig 1Eiii–iv) were observed in the cerulein-with-AOM–treated mice. Semiquantitative average pathology scores in tissue sections of mice treated with saline, AOM, cerulein, and cerulein plus AOM were recorded using light microscopy (Fig 1F). Immunohistochemical analysis detected induced PC-specific PDX1-positive (Fig S1Ai–iv) and SOX9-positive (Fig S1Bi–iv) cells in pancreatic tissue sections of the cerulein-with-AOM–treated mice compared with cerulein-treated mice. The induction of MUC2 expression near the MCN and IPMN region and around the ducts was observed in the pancreas of the cerulein-with-AOM–treated mice (Fig S1Ci–iv). Morphometric quantification of PDX1$^+$, SOX9$^+$, and MUC2$^+$ cells indicated increased expression in cerulein-with-AOM–treated mice (Fig S1D).

### Proteomics analysis detected highly induced eosinophilic granular proteins with several inflammatory and oncogenic proteins in the developed murine model of CP

Next, we used a powerful proteomics approach to identify the mechanistic pathways involved in promoting the CP-associated characteristics observed in PC. Liquid chromatography mass spectrometry (LC–MS) protein profiling of cerulein-with-AOM–treated mice, compared with mice treated with saline or cerulein alone, revealed a hierarchical clustering heat map detecting a total of 2,885 proteins. Among these, a statistically significant (1.5-fold) 131 induced proteins and 50 reduced proteins were found in the cerulein-with-AOM–treated mice compared to mice treated with cerulein alone (Fig 2A). Details of the 2,885 proteins detected by our proteomic analysis are provided in Table S1. The average differences in the individually induced and reduced proteins (fold change, $P < 0.05$) are shown as a volcano plot (Fig 2B). Venn diagram analysis indicated 607 unique induced proteins in the cerulein versus saline, cerulein-with-AOM versus saline, and cerulein-with-AOM versus cerulein mice. In addition, 105, 14, and 328 induced proteins were detected in the cerulein group compared to the saline group, in the cerulein-with-AOM group compared to the saline group, and in the cerulein-with-AOM group compared to the cerulein group, respectively. In addition, details of 328 induced proteins were further analyzed in the cerulein-with-AOM group compared to the cerulein alone group (Fig 2C). The analysis detected 104 common induced inflammatory, profibrotic and oncogenic signature proteins in the saline-normalized cerulein-with-AOM group of mice compared with the cerulein-with-AOM group of mice.

Most importantly, the proteomic analysis detected highly induced eosinophilic granular protein eosinophil peroxidase (EPX) followed by associated macrophages (ARG2 and MRC1) and neutrophils (MPO) (Fig 2D). Similarly, highly induced profibrotic proteins such as lumican (LUM), periostin (POSTIN), and fibroblast growth factor (FGF1) (Fig 2E) and highly induced PC-associated oncogenic proteins like SPRR1A and AKR1B8 (Fig 2F) were observed in mice treated with cerulein plus AOM compared to those treated with cerulein alone or saline. Furthermore, the subcellular localization of the differential proteins in cerulein with AOM versus cerulein groups indicated that they were in the cytoplasm (19.72%), extracellular region (19.72%), plasma membrane (14.79%), mitochondria (13.38%), nucleus (12.68%), and endoplasmic reticulum (6.34%), with the rest of the proteins found in the Golgi apparatus (4.23%), lysosome (2.8%), cytoskeleton, peroxisome, microsome, synapse, and centrosome (Fig 2G). In summary, the proteomic analysis data identified several novel mechanistic proteins involved in promoting chronic pancreatic inflammation that may be involved in the development of some characteristic features associated with PC in the presented CP model.

### Proteomic analysis of the stepwise progression of inflammatory pathway in promoting malignant phenotype in a murine model

Proteomic analysis detected several proinflammatory cellular proteins. Among these, EPX and macrophage-associated protein (MRC1) were highly significantly (several-fold) induced in the pancreas of an inflammation-mediated cerulein-with-AOM–treated experimental model of CP. Thus, we further investigated the mechanistic pathway that regulates macrophage-mediated induction of eosinophilic inflammation in the pancreas. We present evidence that tissue-accumulated macrophages

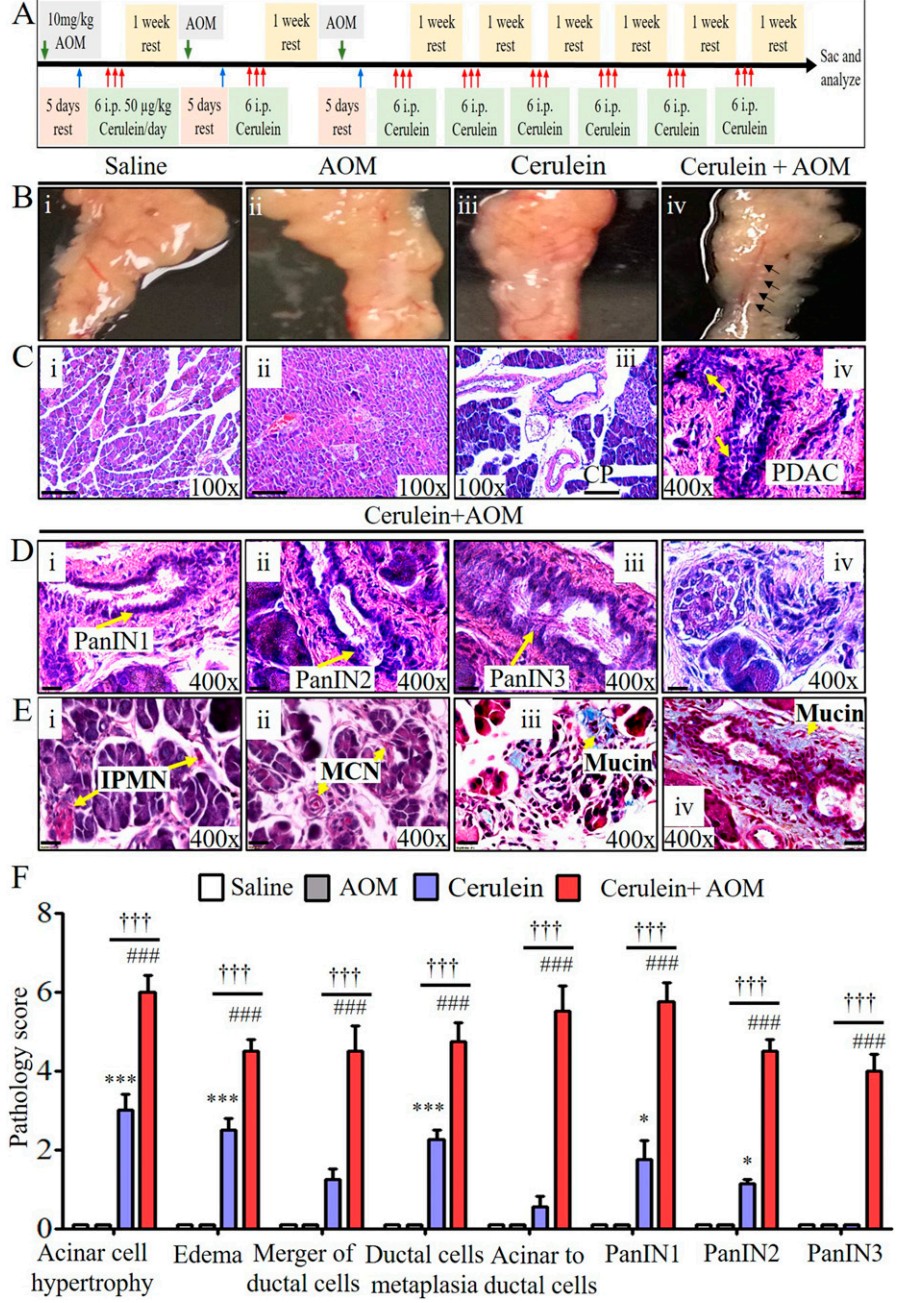

**Figure 1. Cerulein-with-AOM–treated mice develop chronic inflammation-mediated pathological malignant phenotype.**
**(A)** Schematic representation of cerulein with AOM treatment protocol regimen. **(B)** Representative morphological characterization of the pancreas following saline-, AOM-, cerulein-, and cerulein-with-AOM–treated mice (moderate calcified tumor growth indicated by arrows). **(C)** Representative hematoxylin and eosin–stained histological characterization indicating normal acinar cells, ductal cells and islet cells in the pancreas of saline- and AOM-treated mice (Ci-ii). **(Ciii-iv)** Pathological features of malignant phenotype like nuclear abnormalities, loss of polarity, nuclear overcrowding, enlarged nuclei, hyperplasia, and pancreatic duct fusion are visible in the cerulein-with-AOM–treated mice compared to the acinar cell hypertrophy and ductal hyperplasia in cerulein-treated mice (Ciii–iv). **(D)** Representative photomicrographs show the formation of PanIN1, PanIN2, and PanIN3 with periductal stroma and acinar-to-ductal metaplasia (Di–iv). **(E)** Detection of intraductal papillary mucinous neoplasm, mucinous cystic neoplasm, and acinar-to-ductal metaplasia region and periductal mucin accumulation in cerulein with AOM-treated mice (Ei–iv). **(F)** The semi-quantitative pathology scores analysis using light microscopic analysis (F). The data represent the means ± SD, n = 12 mice/group. * or # or †$P < 0.05$, ** or ## or ††$P < 0.001$, *** or ### or †††$P < 0.0001$. Symbols represented as *cerulein versus saline and AOM, #cerulein with AOM versus saline and AOM, and †cerulein with AOM versus cerulein. All photomicrographs are 100× (scale bar 100 μm) and 400× (scale bar 20 μm) of original magnification.

show highly induced activated NLRP3 in the tissue sections of mice treated with cerulein plus AOM compared with mice treated with cerulein alone, AOM with saline, or saline alone (Fig 3A). Increased circulating CD11b[+] macrophages and eosinophils were observed in cerulein-with-AOM–treated mice compared with the cerulein-alone–treated mice, whereas very few macrophages were observed in the saline-treated mice (Fig S2Ei–v). Furthermore, analysis using the combination of anti-CD11b/anti-CD86 and anti-CD11b/anti-CD206 double immunofluorescence we observed that M1 macrophages are highly induced compared with M2 macrophages in the pancreas tissue sections of mice treated with cerulein plus AOM compared with mice treated with cerulein, AOM plus saline, or

saline alone treated mice (Fig S2B–D). The induction of pNLRP3, NLRP3, and NLRP3-regulated caspase-1–induced IL-1β and IL-18 was analyzed by performing Western blot (Fig 3B). IL-18 has been reported to cause eosinophil accumulation in tissues; therefore, we further validated induced levels of IL-18 by performing ELISA in the pancreases of cerulein-with-AOM–treated mice compared with mice treated with cerulein, AOM with saline, or only saline (Fig 3B and C). Our proteomics analysis detected high levels of the eosinophil granular protein (EPX) and macrophage related MRC1; these data are further validated by performing Western blot analysis (Fig 3B). Similarly, we also show the accumulation of tissue eosinophils by performing anti-EPX antibody immunohistochemical analysis. The

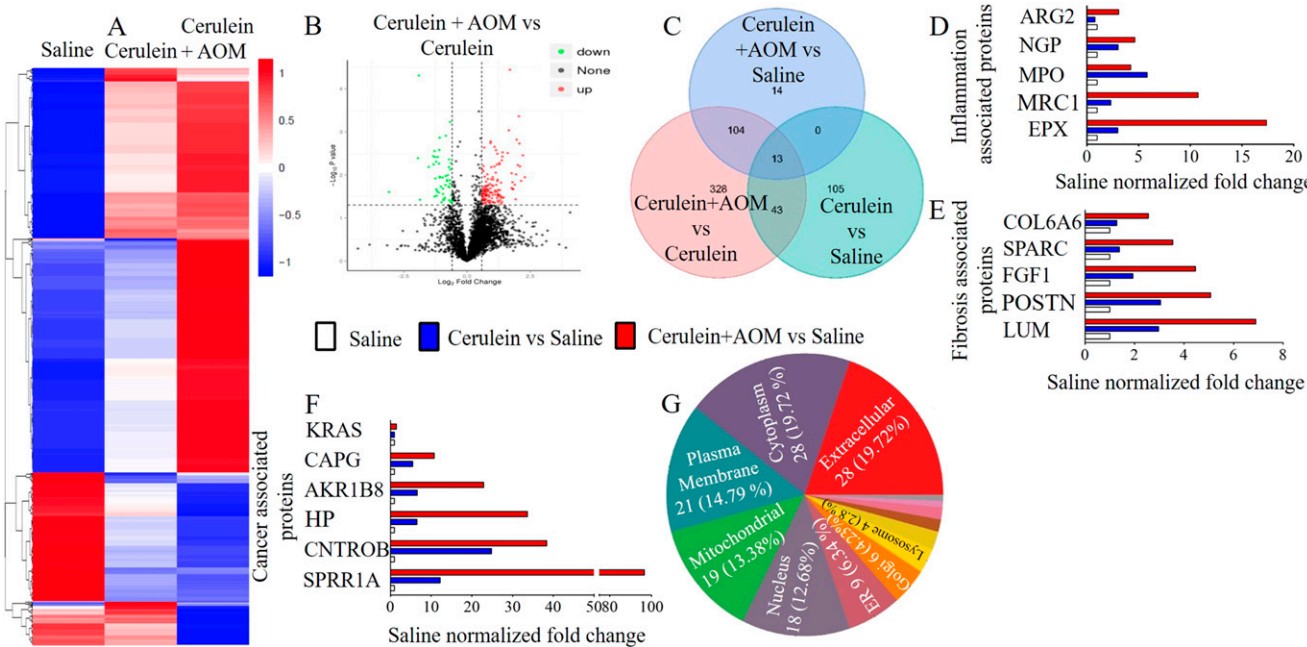

**Figure 2. Mass spectrophotometry proteomics analysis in the murine model of chronic inflammation-mediated malignant pancreatitis.**
**(A)** Heat map of differentially expressed proteins in the pancreases of the saline-, cerulein-, and cerulein-with-AOM–treated mice (red signifies up-regulated and blue down-regulated). **(B)** Volcano plot of differentially expressed protein fold-change expression levels of proteins between all three groups (red dots signify up-regulated, green dots down-regulated). **(C)** Overlapping induced proteins between the groups are shown by Venn diagram. **(D, E, F)** Fold change highly up-regulated proteins associated with inflammatory cells (D), profibrotic proteins (E), and prooncogenic proteins (F). Detection and characterization of subcellular percent localized proteins in the pancreases of mice treated with cerulein plus AOM compared to mice treated with cerulein alone. Data are expressed as means ± SD, n = 3 mice/group.

anti-CCR3/SiglecF+ flow cytometer analysis also detected induced circulating eosinophils in the experimental model of CP. Both analyses detected induced EPX, MRC1 proteins, and intact eosinophils with extracellular EPX-positive granules in mice treated with cerulein plus AOM compared with mice treated with cerulein alone. Very few eosinophils were observed in the saline- or AOM-treated mice (Figs 3Di–iv and S2Ai–iv). Interestingly, degranulated eosinophils and extracellular EPX$^+$ granular proteins were observed around the ADM region (Fig 3Div). The presence of eosinophils in pancreatic tissue sections was further confirmed using anti-MBP (major basic protein) antibody immunostaining (Fig S3Ai–iv). Next, we aimed to understand the mechanism underlying eosinophil accumulation; therefore, we examined the expression of vasoactive intestinal polypeptide (VIP) in the pancreas of our murine model. VIP has been shown to have chemoattractant activity for eosinophils similar to the chemokine eotaxin (Verma et al, 2018). Eotaxins were reported induced in the cerulein-induced CP in mice (Manohar et al, 2018b); whereas, another eosinophils chemoattractant neuropeptide VIP was reported in the human pancreatic cancer (Tang et al, 1997; Moody et al, 2016). Herein, we show that VIP is significantly induced in cerulein-with-AOM–treated mice compared to mice treated with cerulein or saline alone (Fig 3E i–iv). The VIP chemoattractant activity for eosinophils is further demonstrated by in vitro concentration-dependent manner, as well by ex vivo 3D gel analysis (Fig S4Ai–ii and B). The F4/80$^+$, NLRP3$^+$, EPX$^+$, and VIP$^+$ cells were quantified in the tissue sections by performing morphometric analysis (Fig 3F). In addition, double immunofluorescence staining with anti-EPX and anti-VIP antibodies detected eosinophils near the VIP-expressing nerve cells in the area of the ductal cells and the ADM region (Fig S3Bi–iv). These data provide a mechanistic understanding of the role of NLRP3-regulated IL-18-

induced eosinophilic inflammation in the development of several characteristic features observed in CP-associated PC.

## Eosinophilic chronic inflammation promotes pancreatic tissue remodeling and fibrosis

Eosinophils are an established source of TGF-$\beta$; therefore, accumulation of eosinophils may induce TGF-$\beta$-mediated pancreatic fibrosis. Accordingly, we examined the induction of TGF-$\beta$ and the TGF-$\beta$ signaling molecule SMAD4 in the murine model of CP by performing Western blot and immunohistochemical analyses. The immunoblot analysis detected highly induced levels of TGF-$\beta$ and SMAD4 in cerulein-with-AOM–treated mice compared with mice treated with cerulein alone, AOM plus saline, and saline alone (Fig 3G). Immunohistochemical analysis validated the data and showed significantly induced anti-TGF-$\beta^+$ and TGF-$\beta$ signaling molecule anti-SMAD4$^+$ cells in pancreatic tissue sections of mice treated with cerulein plus AOM compared to those treated with cerulein, AOM plus saline, and saline alone (Fig 3Hi–iv and Ii–iv) Furthermore, we also detected more alpha smooth muscle actin ($\alpha$-SMA)+ cells in cerulein-with-AOM–treated mice compared with those treated with cerulein, AOM plus saline, and saline alone (Fig S3Ci–iv). A statistically significant induced number of TGF-$\beta^+$, SMAD4$^+$, and $\alpha$-SMA$^+$ cells were observed by performing morphometric analysis (Figs 3J and S3E). In addition, we also detected induced comparable periductal collagen accumulation by performing Masson's trichrome staining (Fig S3Di–iv) and deposited collagen thickness by performing morphometric quantitative analysis in the cerulein-treated and cerulein-with-AOM–treated mice (Fig S3F).

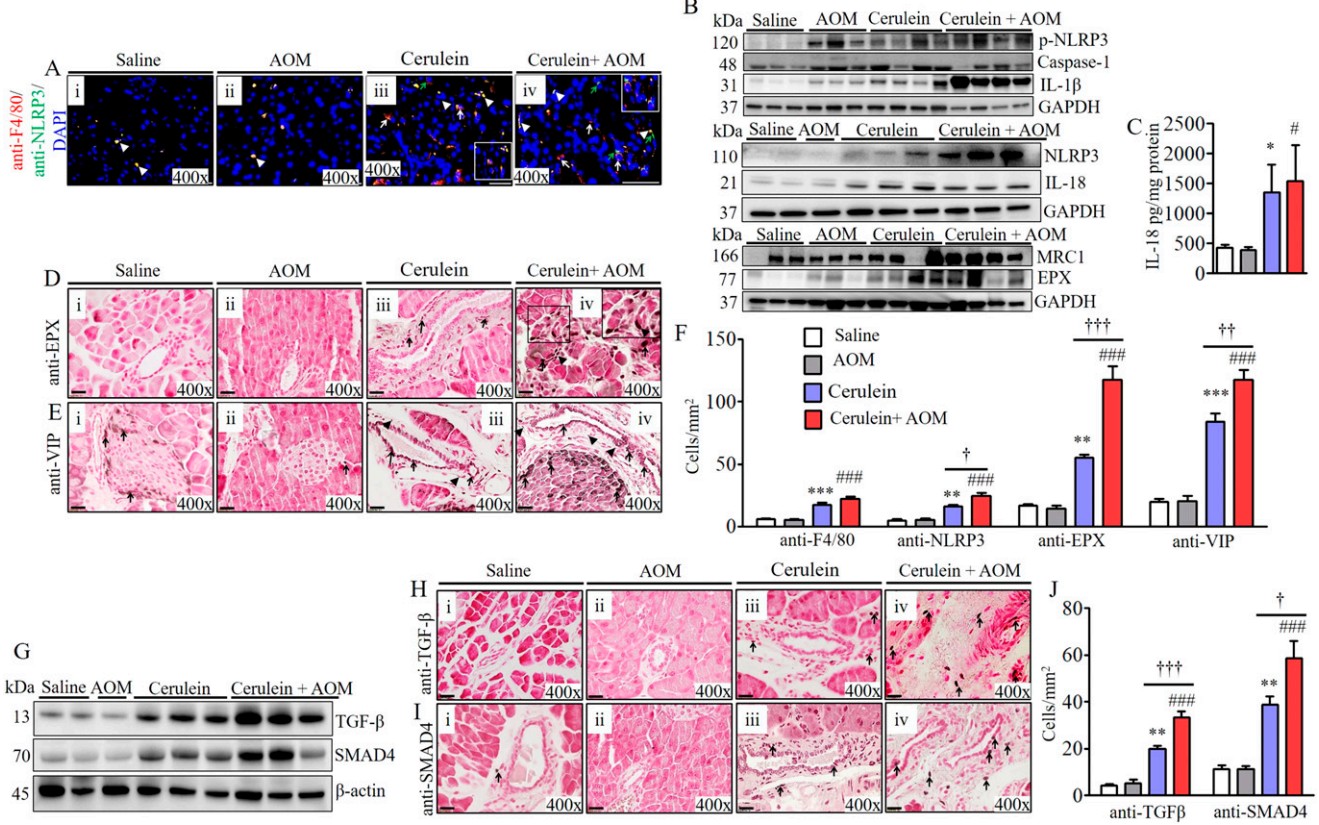

**Figure 3. Mechanistic molecular pathway involved in eosinophilic inflammation-mediated fibrosis in cerulein- and AOM-treated mouse model of chronic pancreatitis.**
**(A)** Immunofluorescence analysis detected induced NLRP3+ (white arrows) and F4/80+ (green arrows) macrophages, and F4/80 and NLRP3 double positive cells (arrow heads) (i–iv). **(B)** Western blot analysis for p-NLRP3, caspase-1, NLRP3, IL-1β, IL-18, MRC-1, and EPX. **(C)** ELISA analysis for IL-18. **(D)** Highly induced EPX+ eosinophils detected in tissue sections by immunohistochemical analysis (i–iv). **(E)** Nerve cells expressing eosinophil chemoattractant protein vasoactive intestinal polypeptide in pancreatic tissue section of mice (i–iv). **(F)** Morphometric quantification analysis for F4/80, NLRP3, EPX, and vasoactive intestinal polypeptide expressed as cells/mm². **(G)** Immunoblot analysis of the profibrotic protein TGF-β and signaling molecule SMAD4. **(H)** Immunohistochemical analysis detected highly induced TGF-β+ cells in pancreatic tissue sections (i–iv). **(I)** Immunohistochemical analysis detected induction of SMAD4-positive cells in pancreatic tissue sections (i–iv). **(J)** Morphometric quantification analysis detected TGF-β and SMAD4 positive cells, expressed as cells/mm². * or # or †$P < 0.05$, ** or ## or ††$P < 0.001$, *** or ### or †††$P < 0.0001$. *Represents cerulein versus saline and AOM, # cerulein with AOM versus saline and AOM, and † cerulein with AOM versus cerulein. Data are expressed as means ± SD, n = 8 mice/group. All photomicrographs shown are 400× (scale bar 20 μm) the original magnification.

## Chronic eosinophilic inflammation induces oncogenic proteins that are linked to the formation of PanINs and ADM in CP-associated PC

Our proteomic analysis detected highly induced small proline rich protein 1A (SPRR1A) and kirsten rat sarcoma viral oncogene homolog (KRAS) in cerulein-with-AOM–treated mice compared to the respective control group of mice including cerulein-treated mice. Therefore, we first validated the induction of SPRR1A and KRAS along with another common oncogenic protein, p53, by performing Western blot analysis. The analysis indeed showed significantly induced SPRR1A, KRAS, p53 in cerulein-with-AOM–treated mice compared with the control groups of mice, including cerulein-treated mice (Fig 4A and B). Furthermore, we examined the location of the cells expressing these induced oncogenic proteins by performing immunohistochemical analysis, which detected induced SPRR1A, KRAS, p53 (Fig 4C–E), transcription termination factor 1 (TTF-1), and vascular

endothelial growth factor (VEGF) (Fig S5B and C) proteins nearby the formation of PanINs, merging of ducts, and ADM. Immunofluorescence staining showed highly induced p53 (green) positive cells in the ductal cells and the ADM region in mice treated with cerulein alone and cerulein with AOM (Fig S5Ai–iv). An immunoglobulin G (IgG) control antibody did not detect any positive cells (Fig S5Av). Statistically significant increases in SPRR1A+, KRAS+, p53+, TTF-1+, and VEGF+ cells were observed by performing morphometric analysis (Figs 4F and S5D). Because eosinophil degranulated proteins are implicated in cell damage, we also examined cell cycle proteins and the phosphorylation of signal transduction molecules in the pancreas. Immunoblot analysis showed induced CDK9 and the inactivation of the cell cycle protein CDKN2A, along with induced phosphorylation of extracellular signal-regulated kinase (ERK), AKT serine/threonine kinase (AKT), and epidermal growth factor receptor (EGFR) in the cerulein- and cerulein-with-AOM–treated mice compared with saline- and AOM-treated mice (Fig 4G).

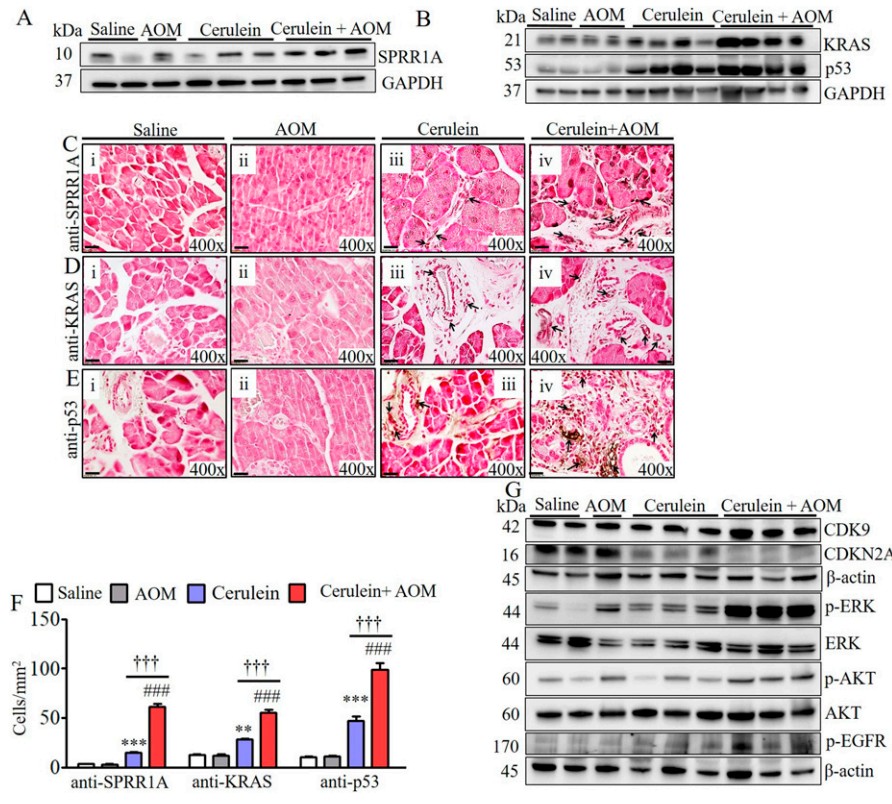

**Figure 4. Detection of induced oncogenic proteins in cerulein-with-AOM–induced murine model of chronic pancreatitis.**
**(A, B, C, D, E)** Validation of proteomic detected induced SPRR1A (A) and KRAS along with p53 (B) by Western blot analysis. SPRR1A, KRAS, and p53 expressed cell detection by performing immunohistochemical analysis (C, D, E). **(F)** Morphometric quantification of SPRR1A-, KRAS-, and p53-positive cells, expressed as cells/mm$^2$ (F). **(G)** Immunoblot analysis of CDK9, CDKN2A, p-ERK, p-AKT, and p-EGFR protein levels (G). Data are expressed as means ± SD, * or # or †$P < 0.05$, ** or ## or ††$P < 0.001$, *** or ###or †††$P < 0.0001$. *Represents cerulein versus saline and AOM, # cerulein with AOM versus saline and AOM, and † cerulein with AOM versus cerulein. n = 6–8 mice/ group. All photomicrographs are shown in original magnification of 400× (scale bar 20 μm).

### Anti–IL-18 neutralization and IL-18 deficiency in CP murine model significantly reduces pancreatic eosinophilia and the development of pathological PC phenotype

Last, we set out to establish a critical role of IL-18–induced eosinophilic inflammation and a therapeutic immune checkpoint in CP-associated PC. We tested the hypothesis that anti–IL-18 immunotherapy may be a novel immune checkpoint to protect the development of several characteristic features that develop in PC. Accordingly, anti–IL-18 immunotherapy was performed on our cerulein-with-AOM–treated mouse model. The anti–IL-18 pretreatment immunotherapy protocol regimen used to test our hypothesis is presented in the schematic diagram in Fig S6A. The data obtained following the anti–IL-18 treatment regimen showed a restoration of several characteristic features of PC including the merging of pancreatic ducts, formation of PanIN1, PanIN2, PanIN3, IPMN, and MCN in cerulein-with-AOM–treated mice compared with the cerulein-alone–treated mice. The anti–IL-18–treated mice even showed improved acinar cell hypertrophy and reduced stroma around the pancreatic ducts (Figs 5Ai–v and S6D). Reduced PDX1-positive cells with improved PanIN1, PanIN2, and PanIN3 formation were observed in cerulein-with-AOM–treated mice also treated with anti–IL-18, compared with induced PDX1-positive cells nearby the PanINs in the pancreatic ducts of cerulein-with-AOM–treated mice without anti–IL-18 (Fig 5Bi–v). Most importantly, the anti–IL-18–pretreated cerulein-plus-AOM–treated mice showed significantly reduced EPX-positive eosinophils with highly improved ADM compared with cerulein-plus-AOM–treated mice without anti–IL-18 (Fig 5Ci–v). Similarly, a reduced immunoreactivity of anti–TGF-β (Fig

5Div–v), anti-MUC2 (Fig 5Eiv–v), anti-KRAS (Fig 5Fiv–v), anti-p53 (Fig 5Giv–v), mucin (Fig S6Biv–v), and collagen accumulation (Fig S6Civ–v) in the tissue sections of the anti–IL-18–pretreated and cerulein-with-AOM–treated mice was observed compared with the cerulein-with-AOM–treated mice without anti–IL-18. Morphometric quantitative statistical analysis of anti-PDX1+, anti-EPX+, anti-TGF-β+, anti-MUC2+, anti-KRAS+, anti-p53+ cells, area of collagen accumulation and several other pancreatic characteristics were performed and presented using a pathology scale (Fig S6E and F). The immunoblot analysis further validated the histological finding of reduced levels of several cell cycle and oncogenic pathway signaling proteins such as TGFβ, SMAD4, KRAS, p53, pERK, p-EGFR, and VEGF in the anti–IL-18–pretreated cerulein-with-AOM–treated mice compared to the those without anti–IL-18 pretreatment (Fig 5H). A similar improved anti-EPX+ eosinophil accumulation, protein expression, and associated improved pathological characteristics such as improved tissue fibrosis and ADM was observed in cerulein-with-AOM–treated IL-18 gene–deficient mice compared with wild-type mice (Fig 6A–G). The IL-18 levels in our experimental model clearly indicate that indeed our efforts neutralize IL-18 and our presented improved CP pathogenesis is associated with IL-18–induced eosinophils accumulation (Fig 5I).

### NLRP3-regulated IL-18–induced eosinophils detected nearby abnormal pancreatic ducts in human PC

Last, we showed that a similar IL-18–induced eosinophil-mediated mechanism is also operational in human PC. We present evidence that eosinophils accumulate near merged ductal cells with PanINs

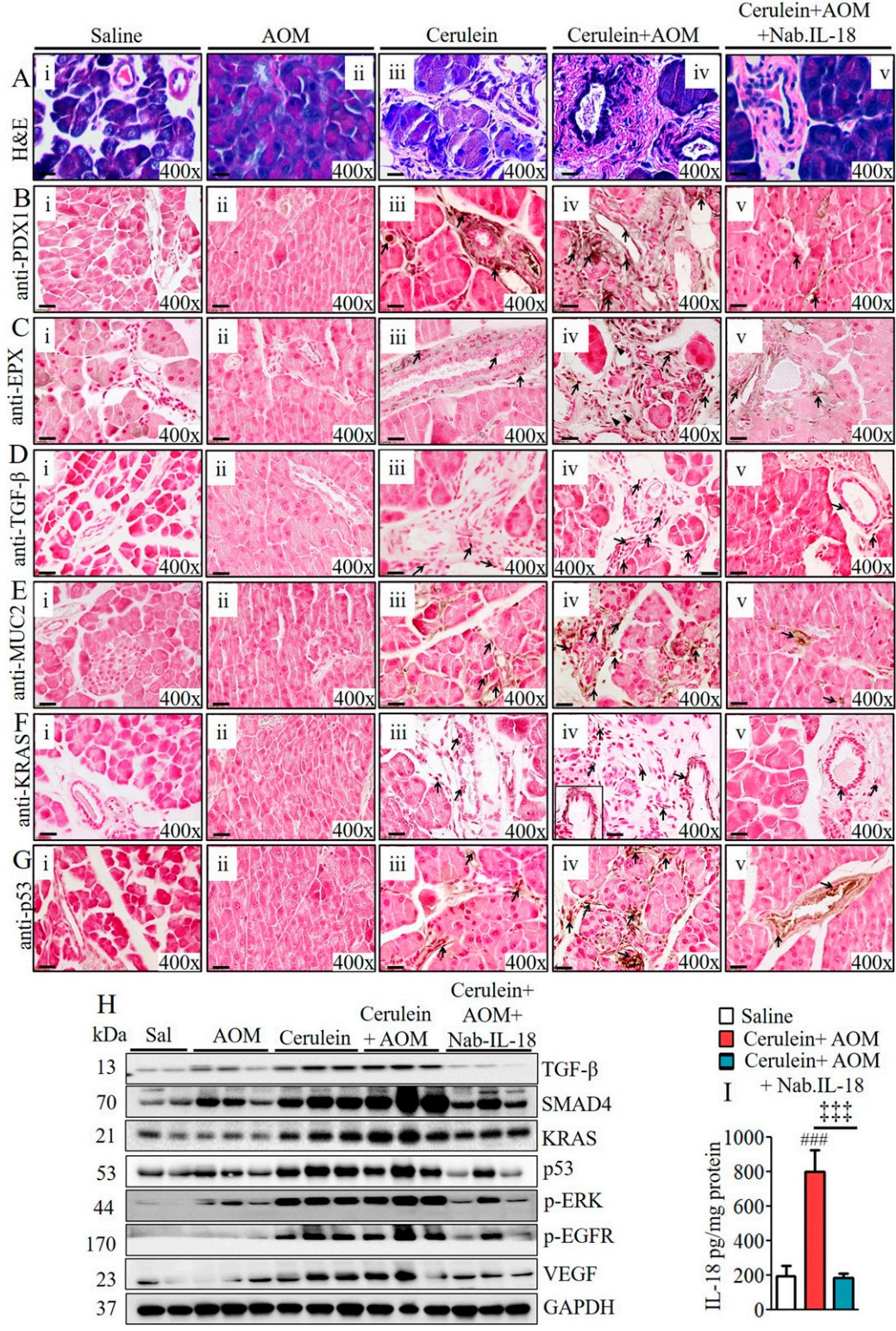

**Figure 5. Anti–IL-18 pretreatment improves malignant pathological phenotype in cerulein- and AOM-treated murine model of chronic pancreatitis.**
**(A)** Representative photomicrographs show improved pancreatic pathology of acinar cell hypertrophy, accumulation of inflammatory cells, ductal hyperplasia, and formation of PanINs and periductal stroma following IL-18 neutralization compared to non-neutralized cerulein-with-AOM–treated murine model of chronic pancreatitis (Aiv–v). **(B, C, D, E, G)** Highly reduced PDX1- (B), EPX- (C), TGF-β- (D), MUC-2- (E), KRAS- (F), and p53- (G) positive cells were observed in IL-18–neutralized compared with non-neutralized cerulein-with-AOM–treated mice. All photomicrographs shown are in original magnification of 400× (scale bar 20 μm). **(H)** Reduced total protein expression levels of TGF-β, SMAD4, KRAS, p53, p-ERK, p-EGFR, and VEGF in IL-18–neutralized compared with non-neutralized cerulein-with-AOM–treated mice (H). **(I)** IL-18 ELISA in pancreatic tissues of saline, cerulein-with-AOM, and cerulein-with-AOM plus IL-18 neutralization treatment (I). Data are expressed as means ± SD, n = 8 mice/group.

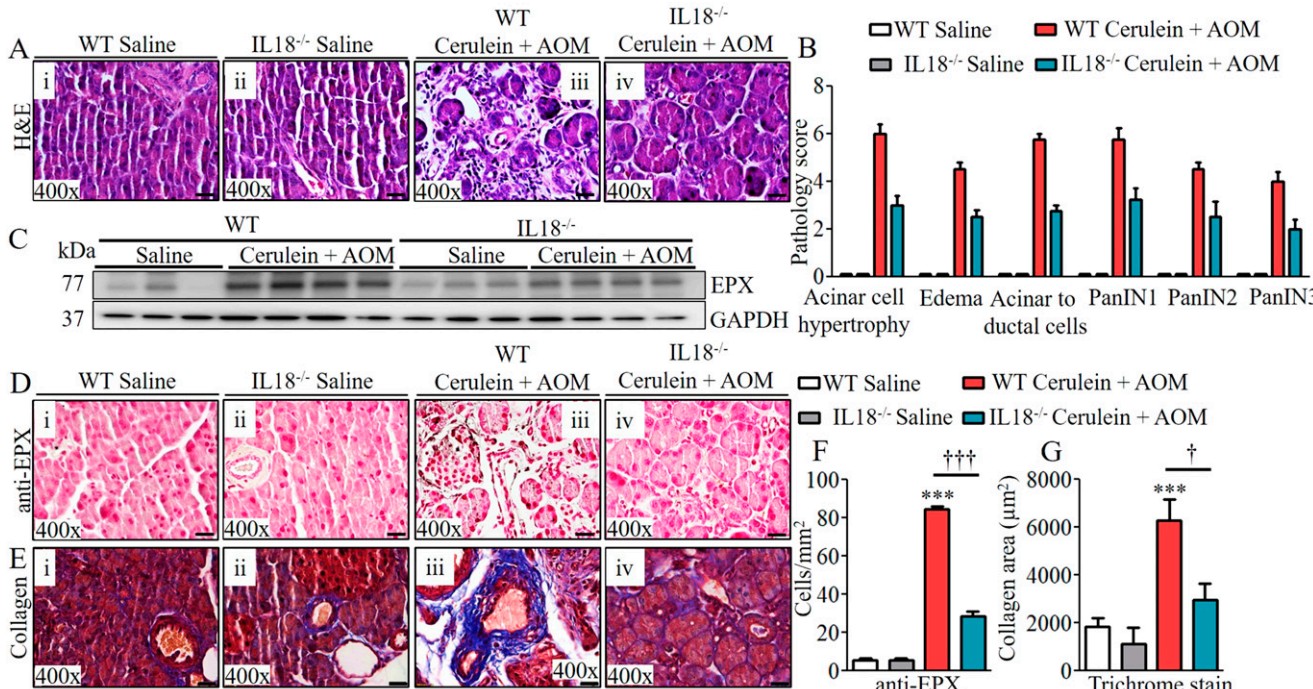

**Figure 6. Improved malignant pathological phenotype detected in cerulein- and AOM-treated IL-18 gene–deficient mice.**
**(A)** Representative photomicrographs show improved pancreatic pathology of acinar cell hypertrophy, accumulation of inflammatory cells, ductal hyperplasia, and formation of PanINs and periductal stroma in IL-18⁻/⁻ mice compared with cerulein-with-AOM–treated murine model of chronic pancreatitis (Ai–iv). **(B, C)** Semi-quantitative pathology scores presented using light microscopic analysis (C). Highly reduced EPX protein expression by immunoblotting in IL-18⁻/⁻ mice compared with cerulein-with-AOM–treated mice. **(D, E)** Immunohistology further confirmed reduced EPX positive cells in IL-18⁻/⁻ mice compared to cerulein-with-AOM–treated murine model of chronic pancreatitis (E), Masson trichrome analysis for collagen staining showed reduced collagen area in IL-18⁻/⁻ mice compared with cerulein-with-AOM–treated mice. **(F, G)** Morphometric analysis shows EPX+ cells and collagen area in IL18⁻/⁻ mice with or without cerulein + AOM treatment. Data are expressed as means ± SD, n = 8 mice/group. All photomicrographs shown are in original magnification of 400× (scale bar 20 µm). †P < 0.05, *** or †††P < 0.0001. *Represents WT cerulein with AOM versus WT saline and AOM, † IL18⁻/⁻ cerulein with AOM versus WT cerulein with AOM.

in biopsy tissue sections of human PC, similar to what we observed in the presented experimental murine model. Normal acinar cells, duct cells, and islet cells were observed in normal benign tumor biopsies (Fig 7Ai–ii). Western blot analysis also showed induced

NLRP3 in the accumulated macrophages after anti-NLRP3 and anti-CD163 double immunofluorescence analysis in PC biopsy compared with benign normal human pancreatic biopsies (Fig 7G and Bi–ii). In addition, anti-EPX antibody immunostaining revealed several EPX⁺

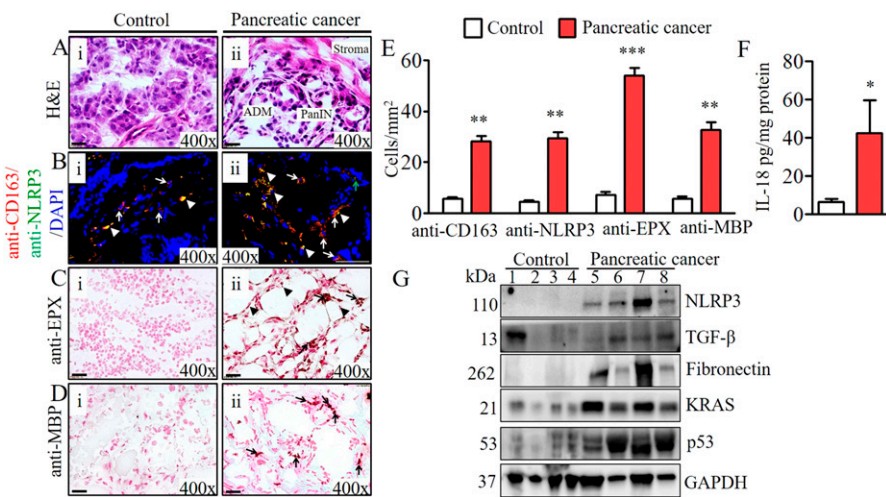

**Figure 7. Human pancreatic malignant tissue mechanistic analysis.**
**(Ai–ii)** Representative hematoxylin and eosin–stained photomicrographs show induced inflammatory cells, merged ductal cells, PanIN formation, and loss of acinar cells compared with the presence of normal acinar cells, ductal cells, and islet cells in healthy pancreas tissue sections (Ai–ii). A representative photomicrograph shows induced NLRP3 (green arrows) in the accumulated macrophages (white arrows) compared with healthy pancreas tissue (Bi–ii). **(Ci–ii, Di–ii)** Detection of intact eosinophils (black arrow) and degranulated extracellular granular proteins (arrow heads) by anti-EPX (Ci–ii) and anti-MBP (Di–ii) in PDAC patient biopsy compared with few eosinophils in normal biopsies. Morphometric quantification of CD163⁺, NLRP3⁺, EPX⁺, and MBP⁺ cells, expressed as cells/mm² (E). **(F)** IL-18 levels in human PDAC compared with the normal (F). **(G)** Levels of NLRP3, TGF-β, fibronectin, KRAS, and p53 in human PDAC compared with the control individuals (G). Data are presented as means ± SD, *P < 0.05, **P < 0.001, ***P < 0.0001. *Represents pancreatic cancer versus control individuals, n = 6–8 human tissues/group. All photomicrographs shown are 400× (scale bar 20 µm) the original magnification.

intact and degranulated eosinophils (Fig 7Ci–ii), further validated by another eosinophilic granular antibody (anti-MBP) in patient biopsies compared with no to very few eosinophils in normal pancreatic biopsies (Fig 7Di–ii). Morphometric quantification detected statistically significant induced CD163+, NLRP3+, EPX+, and MBP+ cells (Fig 7E) and levels of IL-18 (Fig 7F). In addition, the induction of NLRP3, TGF-$\beta$, fibronectin, and oncogenic proteins such as KRAS and p53 confirm that a similar pathway operates to promote inflammation-mediated PC in humans (Fig 7G). These molecular analyses established that the presented chronic inflammation-mediated murine model of pancreatitis-associated PC is novel and may provide a therapeutic immune checkpoint.

# Discussion

Despite major advances in the understanding of pathological characteristics in PC, the factors responsible for the development of these characteristics in CP are not understood. This may be due to the lack of a chronic inflammation-mediated murine model of PC. The current study aimed to reveal the unique molecular events that lead inflammation-mediated CP to progress to PC. In the current study, we established a novel murine model of CP-associated PC by treating mice with a combination of cerulein and AOM. These mice show several pathological features critical to the development of ADM, ductal cell differentiation, and formation of PanINs in CP. AOM is a gene mutating agent previously used to study mechanisms of cancer progression and chemoprevention in dextran sodium sulphate-induced inflammation-mediated colitis (Clapper et al, 2007). Herein, we show the mechanistic events occurring in AOM and cerulein induced inflammation mediated CP. Cerulein is chemically and biologically similar to the human gastrointestinal hormone cholecystokinin-pancreozymin (CCK), which stimulates gastric, biliary, and pancreatic secretion. Cerulein is routinely used to induce acute and CP in rodents (Manohar et al 2018a, 2018b). AOM is chemically similar to Agent Orange, an herbicide used during the Vietnam War and known to promote pancreatic malignancy (Frumkin, 2003; Hertz-Picciotto, et al, 2018). AOM is a potent carcinogen, and has been used to study the underlying mechanisms of inflammation-induced colon cancer in an experimental model of colitis (Clapper et al, 2007). The presented novel murine model of chronic eosinophilic inflammation-induced CP shows most of the characteristics reported in human PC. Gross anatomical observation of the pancreas indicated calcified cellularity with very moderate tumorigenesis. Calcifications in adenocarcinoma can be explained by the occurrence of adenocarcinoma on top of preexisting chronic calcific pancreatitis (Kendig et al, 1966; Haas et al, 1990; Furukawa et al, 1995). Using this unique model and human biopsy samples of human PC, we established the role of the NLRP3-regulated inflammatory cytokine IL-18 in inducing eosinophils and promoting several features associated with the pathogenesis of pancreatic malignancy. We previously reported an increased level of IL-18 in the tissue of experimental pancreatitis (Manohar et al, 2018b), and in the current report, we showed induced NLRP3, IL-18, and eosinophils in a murine model of CP. Several considerable lines of evidence indicate that IL-18 and eosinophils are induced in human CP (Janiak et al, 2015) including

PC (Carbone et al, 2009; Li et al, 2019). However, the direct roles of IL-18 and eosinophils have never been established in the development of pancreatic neoplasms. We provide evidence that accumulated macrophages activate NLRP3-induced IL-18 in the pancreas, which promotes eosinophilic inflammation. IL-18 is capable of generating pathogenic eosinophils from bone marrow progenitors and transforming naïve eosinophils to pathogenic eosinophils (Venkateshaiah et al, 2018; Verma et al, 2019). Herein, we provide stepwise evidence that eosinophil accumulation occurs in a murine model of CP-associated PC. In addition, we also show the mechanism by which vasoactive intestinal peptide (VIP) chemoattracts IL-18–induced eosinophils into the pancreas. Notably, induced VIP has been reported in PC (Tang et al, 1997; Moody et al, 2016) and our in vitro and ex vivo 3D gel eosinophils chemoattraction experiments further provide the significance of VIP role in eosinophils chemoattraction in the pancreas after the induction of CP. We detected the presence of the eosinophilic granular protein eosinophilic peroxidase (EPX), anti–EPX-positive intact eosinophils, and degranulated eosinophilic granular proteins in pancreatic tissue sections of murine models and human PC biopsies, indicating the involvement of eosinophils in the development of several pathological characteristics of PC. Eosinophilic granular proteins are involved in cell damage and proliferation (Venkateshaiah et al, 2018), and the detection of anti–EPX-positive eosinophilic granular proteins near the area of merged ducts, acinar cell hypertrophy, ADM, and PanINs formations suggest the significance of eosinophilic inflammation in CP. An earlier report indicated that EPX was a ligand for the HER2 receptor and a source of TGF-$\beta$ (Kadin et al, 1993) that induced a sustained up-regulation of MUC2 and MUC4, and showed that HER2 was associated with particularly aggressive forms of PC (Lei et al, 1995); therefore, chronic eosinophil activation and the release of EPX granules in tissues contribute to the development of malignancy by activating TGF-$\beta$ and MUC2. These findings strongly support the concept that PanINs form from the differentiation of acinar cells into ductal-like cells as a consequence of eosinophil accumulation and degranulation in a murine model of pancreatitis-associated PC. In addition, eosinophils are a source of TGF-$\beta$ (Kadin et al, 1993) and its signaling via the serine–threonine kinase receptor, which regulates SMAD-2, SMAD-3, SMAD-4, MUC2, and SOX9. The induction of SMADs, MUC2, and SOX9 induced in the presented murine model is critical in the development of the pathogenesis of PC (Burgel et al, 2001; Loktionov, 2019). We also show the induction of MUC2 along with induced collagen, mucin, and fibroinflammatory stroma in the developed eosinophilic inflammation-associated murine model of PC. The Western blot and immunohistochemical analyses on oncogene induction are consistent with the proteomic analysis that detected several other critical oncogenic proteins including ANXA4, LRP1, TAP1, Serpina3, NTAP1, KRAS, SerpinH1, AKR1B8, and SPRR1A in the presented murine model, all of which have been reported as lifetime risk factors for PC in humans. The induced TGF-$\beta$ signaling molecule SMAD4 and the tumor suppressor protein p53 physically interact and jointly regulate the transcription of several TGF-$\beta$ target genes (Cordenonsi et al, 2003). TGF-$\beta$ and receptor tyrosine kinase ligands are pleiotropic cytokines affecting several aspects of cell behavior, ranging from differentiation and proliferation to movement and survival (Schlessinger, 2000; Attisano & Wrana, 2002). KRAS oncogene expression in various settings with additional mutations, including

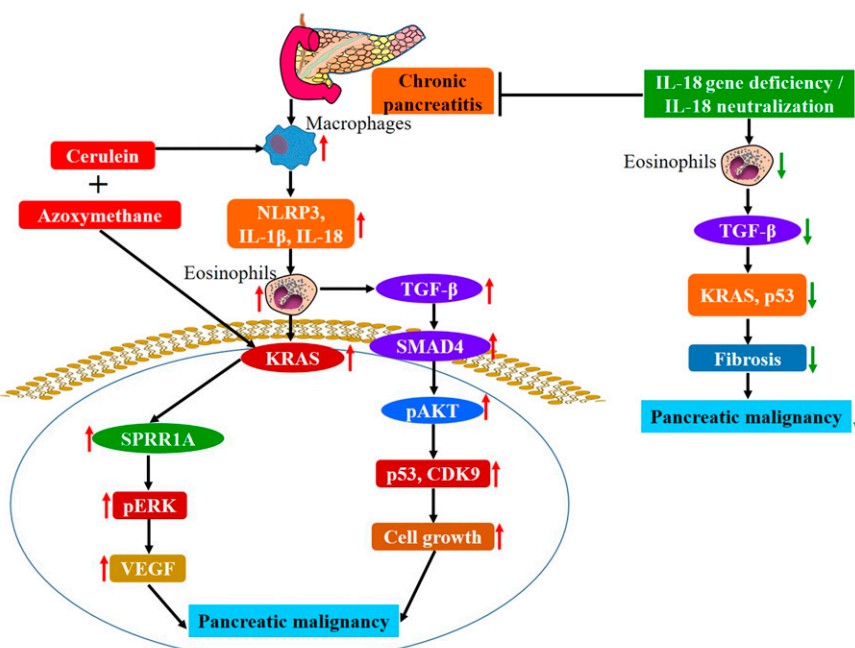

**Figure 8. Mechanism of chronic inflammation-mediated macrophage, NLRP3-IL-1β-IL-18, eosinophils induced TGF-β activation signals KRAS and p53 that lead to pancreatic malignancy in the mice model.**
IL-18 neutralization immunotherapy or IL18⁻/⁻ inhibits eosinophilic inflammation–mediated fibrosis via inhibition of eosinophils, KRAS and p53 and protects pancreatic malignancy.

deletion or inactivation of p53 or TGF-β signaling molecule SMAD4, is significant in tumor development and, in some instances, acquisition of metastatic properties (Aguirre et al, 2003; Tuveson & Hingorani, 2005; Bardeesy et al, 2006). Furthermore, we also show that induced pERK and pAKT play an important role in EGF signaling via its receptor EGFR to promote cell growth and proliferation activating their cytoplasmic kinase domains and resulting in phosphorylation of tyrosine residues (Tsai & Nussinov, 2019) and subsequent recruitment of downstream effectors for initiating various cellular functions (Morandell et al, 2008). This is consistent with our data showing that induced CDK9 and reduced levels of CDKN2A in our mouse model of CP may promote acinar-to-ductal hyperplasia, which combines with reduced levels of CDKN2A to promote cell cycle arrest. Thus, the loss of p16 may represent a common pathway to tumorigenesis. The evidence presented in this study shows that NLRP3-regulated IL-18-induced eosinophilic inflammation may be involved in promoting the pathogenesis of pancreatic neoplasm via the indicated pathways and promotes characteristics similar to human cancer. Most importantly, we present supporting evidence that IL-18 neutralization or IL-18 deficiency in mice restricts the formation of most of the pathological characteristics observed in PC and further relates IL-18–induced pancreatic eosinophilia to promoting CP-mediated development of PC phenotype. We show that both IL-18 gene deficiency and neutralized anti–IL-18 immunotherapy in mice down-regulate profibrotic and oncogenic proteins and pathological characteristic features, such as inhibition of acinar cell hypertrophy, ADM, and proinflammatory stroma, including PanIN1, PanIN2, and PanIN3 formation in a cerulein-with-AOM–induced inflammation-mediated murine model of PC.

Taken together, these current studies provide a novel murine model of chronic inflammation–mediated pancreatitis-associated development of pancreatic neoplasm. We show that activated NLRP3-regulated IL-18 in the accumulated macrophages in the pancreas promote chronic eosinophilic inflammation, which may be the first critical step for the development of pathological characteristic observed in PC in CP. Eosinophil degranulation induces the TGF-β signaling pathway, and the signaling molecule SMAD4 further instigates the oncogenic proteins SPRR1A, KRAS, p53, and MUC2, implicated in the development of pancreatic neoplasm in pancreatitis. The eosinophil granular protein EPX up-regulates TGF-β, KRAS, and MUC2, resulting in collagen and mucin accumulation, ADM, and formation of PanINs, IPMN, and MCN in the pancreas after cerulein-with-AOM treatment in mice. Last, we show that anti–IL-18 immunotherapy is a promising strategy to restrict the eosinophil-mediated development of CP-associated pathological characteristics of PC (Fig 8). In conclusion, our current study establishes that IL-18–induced eosinophilic inflammation mechanistic pathway may be operational in the pathogenesis of CP-induced development of pathological characteristics cancer phenotype that further progress to malignancy. These investigations may also have the potential to provide an immune checkpoint for novel therapeutic strategies to prevent the development of pancreatic malignant neoplasm in CP. In short, we first time present a novel inflammation mediated murine model of CP that requires further investigation to establish the role of IL-18 induced eosinophilic inflammation is critical in promoting CP associated pancreatic malignancy.

# Materials and Methods

### Mice

BALB/c mice (6–8 wk) C57BL/6 and IL-18 gene deficient (IL18⁻/⁻) mice were obtained from Jackson Laboratory and maintained in a

pathogen-free barrier facility. We used only male mice for our study because, as per the literature, CP is more common in males compared with females (Yadav et al, 2011). The Institutional Animal Care and Use Committee approved the animal protocol in accordance with the National Institute of Health guidelines. The experiments were performed according to animal ethical rules and regulations.

## Experimental pancreatitis associated pancreatic adenocarcinoma

CP was induced by repetitive intraperitoneal administration of cerulein and azoxymethane (Sigma-Aldrich) as described in Fig 1A. AOM was given by repetitive intraperitoneal injections (10 mg/kg, one injections/day; three times in the treatment protocol) in 100 μl saline·mice-1. Cerulein was given by repetitive intraperitoneal injections as reported earlier (50 μg/kg, 6 hourly injections/day; 3 d/wk) in 100 μl saline·wk-1·mice (Manohar et al 2018a, 2018b).

## ELISA analysis

ELISA was performed for IL-18, according to the kit supplier protocols, in the saline-, AOM-, cerulein-, and cerulein-with-AOM–treated mouse pancreatic tissue homogenates and human pancreatic tissue homogenates using human/mouse, IL-18 Platinum ELISA kit (Affymetrix, eBiosciences).

## Histopathological analysis

Mouse pancreatic tissue specimens were fixed with 4% paraformaldehyde and embedded in paraffin using standard techniques. The paraffin-embedded sections (5 μm) were stained with hematoxylin and eosin, Masson's trichrome staining, Alcian blue staining, and periodic acid–Schiff (Poly Scientific R&D) staining as described below. Quantification of the pathology score of hematoxylin and eosin stained tissue sections was performed using Olympus cellSens Dimension software and the pathology score was expressed as number. A total of four to five high-power fields in each pancreatic section were evaluated for acinar cell hypertrophy, edema, merging of ducts, acinar to ductal metaplasia, and PanIN1, PanIN2, and PanIN3 positive cell areas.

## Tissue collagen analysis

Collagen staining was performed on tissue sections using Masson's trichrome staining (Poly Scientific R&D) method for the detection of collagen fibers according to the manufacturer's recommendations, and the images were captured using an Olympus BX43 microscope. Morphometric quantitation of collagen was measured using Olympus CellSens Dimension software and the positive area is expressed as square microns.

## Alcian blue staining

Mucin staining was carried out by Alcian blue (Poly Scientific R&D) staining on the tissue sections according to the manufacturer's recommendations, and the images were captured using an Olympus

BX43 microscope. The stained area was quantified using Olympus CellSens Dimension software in square microns.

## Periodic acid–Schiff staining

Mucin staining was carried out by Periodic acid Schiff (Poly Scientific R&D) staining on the tissue sections according to the manufacturer's recommendations, and the images were captured using an Olympus BX43 microscope with Olympus CellSens Dimension software in square microns.

## Immunohistochemistry analysis

Mouse and human pancreatic tissue sections were immunoassayed with inflammatory, fibrotic, and oncogenic proteins as described previously (Manohar et al 2018a, 2018b). Images were captured using an Olympus BX43 microscope, and photomicrographs are presented as original magnification 400×. Quantification of the immunostaining was performed using Olympus CellSens Dimension software and immunohistology staining was expressed as number of positive cells per square millimeter. A total of four to five high-power fields in each pancreatic section were evaluated for respective protein positive cells. The details of antibodies used for immunohistochemistry analysis are listed in Table S2.

## Immunofluorescence analysis

Paraffin-embedded mouse pancreatic tissue sections were deparaffinized and optimal cutting temperature-embedded frozen human pancreatic tissue sections were dehydrated. Antigen retrieval was carried out using the sodium citrate method, blocked with normal goat or donkey serum to reduce nonspecific binding, and incubated with specific primary antibodies followed by secondary antibodies as listed in Table S3. Immunostained sections were mounted with ProLong Gold Antifade Mountant with DAPI (#P36935; Thermo Fisher Scientific). The images were captured using an Olympus BX43 microscope with appropriate filters, and photomicrographs are presented as original magnification 400×.

## Flow cytometry analysis

The total population of the isolated spleen were stained with cell surface-specific antibodies for analysis of eosinophils, and macrophages by flow cytometry. The following antibodies were used for specific antigen analysis: anti-CCR3, and anti-Siglec-F, anti-cd11b, with their respective isotype controls, obtained from eBioscience. The cells were incubated for the specific antigens with the required combination of antibodies at 4°C for 45 min followed by two washes. Flow cytometry analysis was performed using an LSRII (BD Biosciences), Novocyte (ACEA Biosciences), and data were analyzed using FlowJo software.

## Western blot analysis

The pancreas tissues were homogenized and solubilized in Mammalian Protein Extraction Reagent (Thermo Fisher Scientific) containing protease inhibitor cocktail and phosphatase inhibitor (Sigma-Aldrich).

Proteins (20 µg) were resolved on 4–15% MP TGX Gel (Bio-Rad) and transferred to polyvinylidene difluoride (PVDF) membranes (Millipore) (Kandikattu et al, 2021b). The inflammatory, fibrotic, and oncogenic angiogenesis proteins NLRP3, IL-18, TGF-$\beta$, SMAD4, fibronectin, KRAS, p-ERK, ERK, p-AKT, AKT, p53, CDKN2A, CDK9, p-EGFR, and VEGF were detected by Western blotting. GAPDH and $\beta$-actin were used as normalizing controls. The details of antibodies used for immunoblots analysis are listed in Table S4.

## In situ analysis of chemotactic response of VIP to eosinophils

The splenocytes were isolated and stained with anti-CCR3[+] antibody. Cells were placed on presolidified 0.5% agarose gel with VIP, and the movement of eosinophils was photomicrographed using a Bio-Rad microscope. The eosinophils placed on pre-solidified 0.5% agarose gel without VIP served as control eosinophils.

## Eosinophil migration assay

The in vitro chemoattractant behavior of VIP for eosinophils was analyzed using Transwell units (24 wells) with 5-mm porosity polycarbonate filters (Corning, Inc) following the protocol (Verma et al, 2018). The CCR3+ mouse eosinophils were separated by fluorescence-activated cell sorter. The purified mouse eosinophils ($10^5$ cells/well) in Hanks' balanced salt solution, pH 7.2 (Life Technologies) were placed in the upper chamber and different concentrations of recombinant VIP (1, 10, 100, and 500 ng/ml) were added to the lower chamber. The Transwell unit was placed for 4 h in a humidified 95% air–5% $CO_2$ atmosphere at 37°C. After 4 h, media from the lower chamber was centrifuged at 250$g$, and cells were resuspended in phosphate-buffered saline. The number of migrated cells in the lower chamber was counted with a hemocytometer. Each assay was set up in triplicate and repeated three times. Data are expressed as an eosinophil migration index, which is defined as the ratio of the migration of eosinophils in the presence of the chemoattractant VIP, and the migration of eosinophils to the medium control.

## Proteomics analysis and bioinformatics

### Sample preparation and proteomics analysis
Samples were prepared for quantitative proteomic analysis by the addition of 1% SDS and sonication until completely homogenous. The protein concentration was determined using bicinchoninic acid (BCA) protein assay kit (Pierce, Thermo Fisher Scientific). Based on the protein concentration, 50 µg of each sample was prepared for trypsin digestion by reducing the cysteines with DTT followed by alkylation with iodoacetamide. After chloroform–methanol precipitation, each protein pellet was digested with trypsin overnight at 37°C. The digested product was labeled using a tandem mass tag (TMT) pro 16plex Reagent set (Thermo Fisher Scientific Pierce) according to the manufacturer's protocol and stored at –80°C until further use. An equal amount of each TMT-labeled sample was pooled together in a single tube and SepPak purified (Waters) using acidic reversed phase conditions. After drying to completion, an off-line fractionation step was used to reduce the complexity of the sample. The sample was brought up in 100 µl of 20 mM ammonium

hydroxide, pH 10. This mixture was subjected to a basic pH reverse phase chromatography (Dionex U3000; Thermo Fisher Scientific). Briefly, UV monitored at 215 nm for an injection of 100 µl at 0.1 ml/min with a gradient developed from 10 mM ammonium hydroxide, pH 10–100% acetonitrile (ACN) (pH 10) over 90 min. A total of 48 fractions (200 µl each) were collected in a 96-well microplate and recombined in a checkerboard fashion to create 12 "super fractions" (original fractions 1, 13, 25, and 37 became new super fraction #1, original fractions 2, 14, 26, and 38 became new super fraction #2, etc.). The 12 "super fractions" were then run on a Dionex U3000 nanoflow system coupled to a thermo fusion mass spectrometer. Each fraction was subjected to a 90-min chromatographic method using a gradient from 2 to 25% acetonitrile in 0.1% formic acid (ACN/FA) over the course of 65 min, a gradient to 50% ACN/FA for an additional 10 min, a step to 90% ACN/FA for 5 min and a 10-min re-equilibration into 2% ACN/FA. Chromatography was carried out in a "trap-and-load" format using a PicoChip source (New Objective); trap column C18 PepMap 100, 5 $\mu$m, 100 A, and the separation column was EASYSpray C18 PepMap 100, 25 cm, 100 A. The entire run was 0.3 $\mu$l/min flow rate. Electrospray was achieved at 1.8 kV. TMT data acquisition used an MS3 approach for data collection. Survey scans were performed in the Orbitrap using a resolution of 120,000. Data-dependent MS2 scans were performed in the linear ion trap using a collision induced dissociation of 25%. TMT reporter ions were fragmented using high-energy collision dissociation of 60% and detected in the Orbitrap using a resolution of 50,000. This was repeated for a total of three technical replicates. TMT data analysis was performed using Proteome Discoverer 2.3. The three experimental runs of the 12 "super fractions" were merged and searched using SEQUEST HT. The Protein FASTA database was *Mus musculus* (NCBIAV Tax ID = 10090) version 2017-05-05. Static modifications included TMTpro reagents on lysine and N terminus (+304.207), carbamidomethyl on cysteines (= 57.021), and dynamic modification of oxidation of methionine (=15.9949). Parent ion tolerance was 10 ppm, fragment mass tolerance was 0.6 D, and the maximum number of missed cleavages was set to two. Only high scoring peptides were considered using a false discovery rate of 1%.

## Quantitation of protein

The protein quantitation results were statistically analyzed using a $t$ test. The proteins whose quantitation was significantly different between experimental and control groups—$P < 0.05$ and $|log_2FC| > *$ (ratio > * or ratio < * [fold change])—were defined as differentially expressed proteins.

## Functional analysis of protein and differentially expressed proteins

Gene Ontology and InterPro (IPR) functional analysis were conducted using the Interpro scan program against the nonredundant protein database (including Pfam, PRINTS, ProDom, SMART, ProSite, and PANTHER), and the databases of Clusters of Orthologous Groups and Kyoto Encyclopedia of Genes and Genomes were used to analyze the protein family and pathway. Differential protein expressions were used for volcanic map analysis, cluster heat map analysis, and enrichment analysis of Gene Ontology, IPR, and Kyoto

Encyclopedia of Genes and Genomes. The probable protein–protein interactions were predicted using the STRING-db server (http://string.embl.de/).

### Analysis of IL-18 neutralization antibody-treated and IL18$^{-/-}$ pancreas tissues of cerulein- and azoxymethane-challenged mouse model for inflammation, fibrosis, and oncogenic markers

Furthermore, IL-18 neutralization antibody (200 μg/mouse) was administered to mice as described in Fig S5A, a day before each cerulein and azoxymethane (AOM) treatment. CP was induced by repetitive intraperitoneal administration of cerulein and AOM as described in Fig 1A. AOM was given by repetitive intraperitoneal injections (10 mg/kg, one injections/day; three times in the treatment protocol) in 100 μl saline·mice-1. Cerulein was given by repetitive intraperitoneal injections as reported earlier (50 μg/kg, 6 hourly injections/day; 3 d/wk) in 100 μl saline·wk-1 mice-1. In brief, the treatment protocol is AOM on day 1 followed by 5 d rest and six intraperitoneal cerulein injections on days 7, 9, and 11 with a follow-up rest per week and the schedule was repeated for up to 8 wk. In another set of experiments, mice were treated with AOM on day 1 followed by 5 d rest and six intraperitoneal cerulein injections on days 7, 9, and 11 with a follow-up rest per a week and the schedule was repeated for up to three times and after this treatment regime mouse was rested for a week and further treated with six intraperitoneal cerulein injections per day on alternate days for five times and rested for a week, and this treatment regime was continued for four more times. Mice were sacrificed 1 wk after the cerulein injections after eight treatment periods, and tissue was immediately frozen in liquid nitrogen and stored at –80°C until used and tissues also fixed in 4% buffered formalin for histology. The therapeutic effects of IL-18 neutralization antibody treated and IL-18$^{-/-}$ pancreas tissues of cerulein with azoxymethane challenged mouse were analyzed for inflammation, fibrosis, and oncogenic markers by immunoblot and immunohistology analysis as described above.

### Analysis of human PC tissues

Human pancreatic tissues were analyzed by performing anti-MBP and anti-EPX immunostaining, anti-NLRP3 immunofluorescence, ELISA, and immunoblot analysis. The details of human pancreas biopsies are listed in Table S5.

### Statistical analysis

All data were analyzed using GraphPad Prism 5.0 software (GraphPad). Two-tailed unpaired *t* test was used for calculating the statistically significant differences between the means of two independent groups. One-way ANOVA followed by Tukey's post hoc test was used for calculating the statistically significant differences between the means of four or more independent groups.

## Supplementary Information

## Acknowledgements

Dr A Mishra is the Endowed Schlieder Chair; therefore, the authors thank the Schlieder Educational Foundation for its support. The authors also recognize the partial financial support of National Institute of Health (NIH) grant R01 AI080581 (A Mishra) and Tulane University Dean Funds (S Upparahalli Venkateshaiah). Additionally, the authors thank Dr Eric Flemington and Tulane Cancer Center for financial assistance for proteomics analysis. The authors are also thankful to Ms Loula Burton, editor for the Office of Research Proposal Development, Tulane University, for the proofreading and editing of the manuscript.

## Author Contributions

HK Kandikattu: data curation, formal analysis, investigation, methodology, and writing—original draft.
M Manohar: methodology and investigation.
AK Verma: methodology and validation.
S Kumar: methodology.
CS Yadavalli: methodology, review, and editing.
SU Venkateshaiah: methodology, review, and editing.
A Mishra: conceptualization, supervision, funding acquisition, visualization, project administration, and writing—review and editing.

## Conflict of Interest Statement

The authors declare that they have no conflict of interest.

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
