## [Reviewer comments · Life Science Alliance]

Life Science Alliance

Macrophages-induced IL-18 mediated eosinophilia promotes characteristics of pancreatic malignancy

Hemanth Kumar Kandikattu, Murli Manohar, Alok Kumar Verma, Sandeep Kumar, Chandra Sekhar Yadavalli, Sathisha Venkateshaiah, and Anil Mishra

DOI: <https://doi.org/10.26508/lsa.202000979>

Corresponding author(s): Anil Mishra, Tulane School of Medicine

Review Timeline:

Submission Date:	2020-12-08
Editorial Decision:	2021-03-12
Revision Received:	2021-04-30
Editorial Decision:	2021-06-07
Revision Received:	2021-06-11
Accepted:	2021-06-14

Transaction Report:

March 12, 2021

Re: Life Science Alliance manuscript #LSA-2020-00979-T

Prof. Anil Mishra
Tulane School of Medicine
Tulane Eosinophilic Disorder Center, Medicine/Pulmonary
Medicine Pulmonary Diseases
New Orleans, LA 70131

Dear Dr. Mishra,

Thank you for submitting your manuscript entitled "Macrophages-induced IL-18 mediated eosinophilia promotes characteristics of pancreatic malignancy" to Life Science Alliance. The manuscript was assessed by expert reviewers, whose comments are appended to this letter.

We apologize for this extended delay in getting back to you. As you will note from the reviewers' comments below, while the reviewers are interested in these findings, they have raised a number of concerns that should be addressed prior to further consideration of this study at LSA. We, thus, encourage you to submit a revised version of the manuscript to us that addresses all of the reviewers' points. Reviewer 3's pts 1 and 2 can be addressed by text changes.

Thank you for this interesting contribution to Life Science Alliance. We are looking forward to receiving your revised manuscript.

Sincerely,

Shachi Bhatt, Ph.D.

Executive Editor

Life Science Alliance

<https://www.lsjournal.org/>

Interested in an editorial career? EMBO Solutions is hiring a Scientific Editor to join the international Life Science Alliance team. Find out more here -

https://www.embo.org/documents/jobs/Vacancy_Notice_Scientific_editor_LSA.pdf

B. MANUSCRIPT ORGANIZATION AND FORMATTING:

Reviewer #1 (Comments to the Authors (Required)):

The manuscript "Macrophages-induced IL-18 mediated eosinophilia promotes characteristics of pancreatic malignancy" of Kandikattu et al presents interesting data of in vivo analysis of a new mouse model of chronic pancreatitis developing manifestation of pancreatic cancer. The authors claim that IL-18 produced by macrophages can induce eosinophilia and leading to development of

PDAC. I think the report is important especially in context of the establishment of this new murine model. However, I suggest to perform additional experiments to make a conclusion stronger. I believe that this work can be delivered to scientific community after substantial improvements.

Specific comments:

1. The expression of proinflammatory proteins (i.e. EPX and MRC1) detected by proteomic analysis should be validated with Western blot
2. The presence of macrophages and eosinophils in tissues should be validated with FACS analysis. It is interesting to know which phenotype - M1 or M2 do the macrophages possess.
3. Induction of CP should be done also in IL-18 knocked-out mice to verify the evolvement of IL-18 in the eosinophilia development.

Reviewer #2 (Comments to the Authors (Required)):

Based on previous work on the role of eosinophiles in chronic pancreatitis, authors propose a novel PDAC model based on repetitive administration of AOM and cerulein. The model is reminiscent of models of chronic inflammatory bowel disease and colon cancer. Authors suggest a role of macrophage-dependent Nlrp3-mediated IL-18, based on elevated IL-18 levels and a beneficial effect of anti-IL-18 therapy on inflammatory and neoplastic markers. Authors propose a mechanism involving VIP-guided infiltration of IL-18-induced eosinophils.

Major concerns:

1. Nlrp3 biologically activates not only IL-18 but also IL-1beta. Data on the role of IL-1beta in the model presented here are absolutely mandatory.
2. IL-18 is believed to be constitutively expressed in a range of tissues, partly by non-canonical pathways independent of Nlrp3. Concomitant increase of Nlrp3 expression and IL-18 expression as in figure 3 is indicative, however, Nlrp3 activation has to be characterised further. Again, IL-1beta levels might be helpful.
3. IL-18 neutralization should be validated by measuring IL-18 levels.
4. The findings on VIP-mediated eosinophilia infiltration seem rather weak (colocalization). Experiments using silencing techniques or knockout mice might be applied.

Minor concerns:

There are unnecessary spelling and grammar mistakes throughout the manuscript, e.g. „316 Prolong chronic pancreatic inflammation...“

Reviewer #3 (Comments to the Authors (Required)):

The current study by Kandikattu et al., addresses that relationship between pancreatic inflammation, eosinophil infiltration and function of cytokine IL-18 in propensity for transformation to pancreatic malignancy. The authors generated a mouse model of chemically induced pancreatitis combined with a mutagen azoxymethane (AOM) in order to study how inflammation affects eosinophil recruitment and function. Overall, there are some correlative findings that suggest

increased presence of macrophages and eosinophils, together with increased propensity for neoplastic transformation. The mechanistic links highlighted in the abstract are unfortunately not established, the connection to eosinophils is somewhat tangential, and most aspects of the work simply show that there is increased inflammation upon application of caerulein. IL-18 aspect is interesting, but again, it is unclear how it relates to proposed mechanism. In the current form this work unfortunately provides a limited advance to the field.

Major points

1. The choice of model to study inflammation-associated pancreatic neoplasia is not rationalized. There are established models in the field, it is not clear how Caerulein/AOM model induces neoplasia in this case, and the relevance to human disease is not stated.
2. Conclusions are derived based on correlative data, and, therefore, remain unproven. Statement on page 11 'Chronic eosinophilic inflammation induces oncogenic proteins' is completely inaccurate, given the data presented. This is just one example, but similar occurrences are through out the draft. Most of the proposed mechanistic connections appear out of nowhere, with no clear rationale.
3. Samples of human pathology are not clear; I cannot see most of the reported structures in the histology image.

Minor points

1. The written document could be edited for clarity.
2. The method for generating semi-quantitative pathology score (Figure1) is not described.
3. In Figure 3D, there is nothing to indicate that this is an ADM area or nerve cell co-localization.
4. Rationale for investigating VIP is unclear
5. The role and significance of using AOM is unclear. Most findings are also in C alone.
6. Figure 4H, WB data does not correspond to conclusions in text.
7. P53 IF (figure S3) is not nuclear as it should be.

Shachi Bhatt, Ph.D.
Executive Editor
Life Science Alliance

Dear Dr. Bhatt,

Re: Response to reviewers for the Life Science Alliance manuscript #LSA-2020-00979-T

We thank you and all the reviewers for the favorable review of our manuscript. The review of our manuscript indicates that all reviewers appreciated the data presented in the study with some concerns and suggestions to further improve the current study. Accordingly, we modified our manuscript and included some new required data and the explanation to each concern. Please find below our point-by-point responses below and for the review convenience, all changes in the manuscript text are indicated in **bold**.

Response to Reviewer #1 Comments:

We thank reviewer #1 for acknowledging that study have interesting data of in vivo analysis of a new mouse model of chronic pancreatitis developing manifestation of pancreatic cancer. The reviewer suggested few additional experiments to establish the presented experimental model and to make a conclusion stronger. Accordingly, we added new data in the manuscript as per the reviewer's suggestion. Our responses for each suggestion or concerns are as follows.

1. The expression of proinflammatory proteins (EPX and MRC1) detected by proteomic analysis should be validated with Western blot.

As per the reviewer's suggestion, we now included the Western blot data of EPX and MRC1 in all group of mice to validate the presented induced proteomic data in Figure # 3B and details of the experiments are provided in the manuscript text.

2. The presence of macrophages and eosinophils in tissues should be validated with FACS analysis. It is interesting to know which phenotype - M1 or M2 do the macrophages possess.

According to reviver's suggestion, first we performed double immunofluorescence staining with the combination of anti-CD11b/anti-CD86 and anti-CD11b/anti-CD206 for pancreas specific M1 and M2 macrophages on tissue sections, which showed highly induced M1 compare to M2 accumulated macrophage in the experimental model of CP associated phenotype development of pancreatic cancer (Supplementary Figure # 2B&C).

Additionally, since it was technically difficult to do the pancreas accumulated macrophages by FACS analysis; therefore, we examined circulated eosinophils and macrophages using splenocyte and presented the data in Supplementary figure #2 A and E, and details of the experiments are provided in the manuscript text.

3. Induction of CP should be done also in IL-18 knocked-out mice to verify the devolvement of IL-18 in the eosinophilia development.

We earlier performed IL-18 gene-deficient mice data in our model; but not included as we presented the IL-18 neutralization data in the manuscript. However, because reviewer ask to present this data; therefore, as per the suggestion now including IL-18 gene-deficient mice data following saline, AOM, cerulein and cerulein with AOM treatment in additionally included Figure #6. The presented data further verify improved eosinophilia and associated pancreatic cancer phenotype in IL-18 gene-deficient mice following cerulein with AOM treated murine model CP. The details of the experiments are provided in the manuscript text.

Response to Reviewer #2 Comments.

We thank reviewer for favorable review with several concerns on the presented data. Herein, we now address below all the concern point-by-point. Each explanation and additions of the concerns are now included in the revised manuscript text.

1. Nlrp3 biologically activates not only IL-18 but also IL-1beta. Data on the role of IL-1beta in the model presented here are absolutely mandatory.

As per the reviewer's concern we now included the data of IL-1beta. in the Figure # 3B. Since our study was focused on the role of eosinophils and IL-1beta is not involved in this process; therefore, we earlier not presented this data.

2. IL-18 is believed to be constitutively expressed in a range of tissues, partly by non-canonical pathways independent of Nlrp3. Concomitant increase of Nlrp3 expression and IL-18 expression as in figure 3 is indicative, however, Nlrp3 activation has to be characterized further. Again, IL-1beta levels might be helpful.

We further characterize the activation of NLRP3 and included the pNLRP3 and Caspase-1 involved in NLRP3 activation pathway. The data is presented in the Figure # 3B.

3. IL-18 neutralization should be validated by measuring IL-18 levels.

We now present the IL-18 ELISA data to confirm that IL-18 is indeed neutralized in our mouse model, the data is presented in Figure 5I.

4. The findings on VIP-mediated eosinophilia infiltration seem rather weak (colocalization). Experiments using silencing techniques or knockout mice might be applied.

VIP knockout mice are commercially not available, we earlier tried to obtain; but were not successful. The silencing techniques is a good alternative; however please note that our model require almost 5 months to generate this data and standardization of experiments will take additional time. We have these experiments plan for future studies once we hope that reviewer will understand our limitations.

However, as per the reviewer's suggestion, we alternatively present the evidence on the role of concentration dependent VIP in chemoattracting eosinophils via *in vitro* culture technique and ex vivo 3D gel analysis. These data are now presented in Supplementary Figure #4 and the details are included in the text of manuscript. Hope the reviewer will appreciate our efforts to address the issue.

Minor concerns:

There are unnecessary spelling and grammar mistakes throughout the manuscript, e.g. „316 Prolong chronic pancreatic inflammation..."

We apologies for the spelling and grammar errors and now corrected text of discussion section in the manuscript.

Response to Reviewer #3 Comments.

We thank reviewer for the review of our manuscript, we agree with the reviewer that our presented study have indirect evidences of IL-18 induced eosinophils correlation to the CP mediated development of phenotype of pancreatic cancer. Herein, we mechanistically show the deleting or neutralizing the IL-18 gene and protein, respectively reduces the eosinophils accumulation as well and improved the pathological phenotype characteristic of pancreatic cancer in our experimental model, which are the novel finding. We agree that it needs further investigation to directly correlate eosinophils to the disease pathogenesis and we are aggressively working on the project. We hope that our current manuscript will draw the attention of several investigators and readers of your journal on the significance of IL-18-induced accumulated eosinophils in CP associated development of cancer phenotype.

Notably, the direct role of eosinophils in the development of CP associated PDAC is yet not properly investigated. Even though, both in CP and pancreatic cancer eosinophils accumulation is reported in patients. Our responses to each concerns of reviewer #3 is now presented below and addressed in the revised manuscript text.

1. The choice of model to study inflammation-associated pancreatic neoplasia is not rationalized. There are established models in the field, it is not clear how Caerulein/AOM model induces neoplasia in this case, and the relevance to human disease is not stated.

Cerulein is chemically and biologically similar to the human gastrointestinal hormones cholecystokinin-pancreozymin (CCK) that stimulates gastric, biliary, and pancreatic secretion. Several reports show cerulein is routinely used to induce acute and chronic pancreatitis in rodents, similar to human (Manohar et al., 2018a; Manohar et al., 2018b). Azoxymethane (AOM) is chemically similar to Agent Orange, an herbicide used during the Vietnam War and notoriously known to promote pancreatic malignancy in service personnel and citizens (Fallon et al., 1994; Frumkin, 2003; National Academies of Sciences and Medicine, 2018). AOM is a potent carcinogen, and earlier used to study the underlying mechanisms of inflammation-induced colon cancer in experimental model of colitis that mimic human colon cancer. Accordingly, we examine whether cerulein with AOM administration induces inflammation mediated chronic pancreatitis (CP)-associated development of malignant phenotype in the pancreas of mice to understand the mechanism that promote CP associated PDAC in human. This information is now included in the manuscript on page 15.

2. Conclusions are derived based on correlative data, and, therefore, remain unproven. Statement on page 11 'Chronic eosinophilic inflammation induces oncogenic proteins' is completely inaccurate, given the data presented. This is just one example, but similar occurrences are through out the draft. Most of the proposed mechanistic connections appear out of nowhere, with no clear rationale.

We corrected the study conclusion and the sentences throughout in the manuscript by stating that “based on our presented data our study indicates that IL-18 induced eosinophilic inflammation may be responsible for inducing oncogenic proteins and pathological phenotype observed in experimental model, which needs more investigation using the presented novel model of CP.”

3. Samples of human pathology are not clear; I cannot see most of the reported structures in the histology image.

We have changed the human pathology photomicrograph that show much clear structures in the histological images. Please note that these are the frozen sample and the histological characteristics will be little different than the formalin fix sections.

Minor points

1. The written document could be edited for clarity.

We edited the manuscript for more clarity.

2. The method for generating semi-quantitative pathology score (Figure1) is not described.

Methods are now provided for semiquantitative pathological score on Supplementary methods.

3. In Figure 3D, there is nothing to indicate that this is an ADM area or nerve cell co-localization.

ADM area in figure D is now marked.

4. Rationale for investigating VIP is unclear.

Rationale to investigate VIP is provided on line # 205-209 as well in the discussion section on line #366-369.

5. The role and significance of using AOM is unclear. Most findings are also in C alone.

The significance of using AOM is provided in the discussion section

6. Figure 4H, WB data does not correspond to conclusions in text.

Figure 4H data, is now discussed in discussion section of manuscript text.

7. P53 IF (figure S3) is not nuclear as it should be.

P53 IF indicates p53 expression in DAPI stained (blue) nucleus of ductal cells.

We hope that the reviewers will appreciate our efforts and agree with presented new data and explanations are informative and now the manuscript is suitable for the publication in the Journal of Life Science Alliance .

Thanks

Sincerely,

Anil Mishra

Anil Mishra, PhD

June 7, 2021

RE: Life Science Alliance Manuscript #LSA-2020-00979-TR

Prof. Anil Mishra
Tulane School of Medicine
Tulane Eosinophilic Disorder Center, Medicine/Pulmonary
Medicine Pulmonary Diseases
New Orleans, LA 70112

Dear Dr. Mishra,

Thank you for submitting your revised manuscript entitled "Macrophages-induced IL-18 mediated eosinophilia promotes characteristics of pancreatic malignancy". We would be happy to publish your paper in Life Science Alliance pending final revisions necessary to meet our formatting guidelines.

- please consult our manuscript preparation guidelines <https://www.life-science-alliance.org/manuscript-prep> and make sure your manuscript sections are in the correct order
- please upload your supplementary figures as single files
- please make sure the author order in your manuscript and our system match and that there is no name discrepancy of all Authors between the manuscript file and the system (the names of the co-authors must be the same in the manuscript text and in the system)
- please add ORCID ID for the corresponding author-you should have received instructions on how to do so
- please add your supplementary figure and table legends to the main manuscript text after the main figure legends
- please add callouts for Table S3 to the main manuscript text
- please incorporate the Supplemental Methods into the main Materials & Methods section

FIGURE CHECKS:

- Please include scale bars for all micrograph images, and indicate the size of the scale bar in the Figure Legend
- Please indicate molecular weight next to each protein blot

A. FINAL FILES:

B. MANUSCRIPT ORGANIZATION AND FORMATTING:

Sincerely,

Reviewer #1 (Comments to the Authors (Required)):

Many thanks to the authors providing an excellent revision

Reviewer #2 (Comments to the Authors (Required)):

Authors provide evidence that NLRP3-mediated IL-18 promotes eosinophilic inflammation and associated malignancy in a pancreatitis model. The manuscript now contains data on IL-1beta and on efficacy of anti-IL-18 therapy. Authors could not demonstrate VIP-silencing data mentioning that these experiments were beyond the scope of this manuscript. However, authors present in vitro studies indicating a chemoattractant effect of VIP on eosinophils. In summary, my remarks have been adequately addressed. Despite some mechanistic weaknesses, I find the connection between NLRP3-mediated IL-18, eosinophilia and inflammation-induced malignancy interesting, novel and worthy to be published in this journal.

June 14, 2021

RE: Life Science Alliance Manuscript #LSA-2020-00979-TRR

Prof. Anil Mishra
Tulane School of Medicine
Tulane Eosinophilic Disorder Center, Medicine/Pulmonary
Medicine Pulmonary Diseases
New Orleans, LA 70112

Dear Dr. Mishra,

Thank you for submitting your Research Article entitled "Macrophages-induced IL-18 mediated eosinophilia promotes characteristics of pancreatic malignancy". It is a pleasure to let you know that your manuscript is now accepted for publication in Life Science Alliance. Congratulations on this interesting work.

*****IMPORTANT:** If you will be unreachable at any time, please provide us with the email address of an alternate author. Failure to respond to routine queries may lead to unavoidable delays in publication.*******

DISTRIBUTION OF MATERIALS:

Again, congratulations on a very nice paper. I hope you found the review process to be constructive and are pleased with how the manuscript was handled editorially. We look forward to future exciting submissions from your lab.

Sincerely,
